# The Business of DNA Nanotechnology: Commercialization of Origami and Other Technologies

**DOI:** 10.3390/molecules25020377

**Published:** 2020-01-16

**Authors:** Katherine E. Dunn

**Affiliations:** School of Engineering, Institute for Bioengineering, University of Edinburgh, The King’s Buildings, Edinburgh, Scotland EH9 3DW, UK; k.dunn@ed.ac.uk

**Keywords:** DNA nanotechnology, DNA origami, patents, commercialization

## Abstract

It is often argued that DNA nanotechnology has a multitude of possible applications. However, despite great advances in the understanding of the fundamental principles of the field, to date, there has been comparatively little commercial activity. Analysis of patent applications and company case studies suggests that this is now starting to change. The number of patent application filings is increasing, and new companies are being formed to exploit technologies based on nanoscale structures and devices made from DNA. There are parallels between the commercial developments in this field and those observed in other areas of innovation. Further commercialization is expected and new players will emerge.

## 1. Introduction

The proposed applications of DNA nanotechnology [1] are wide-ranging, from medical technologies such as targeted drug delivery [2,3] to nanolithography [4,5], to the construction of molecular breadboards for enzymes [6] or polymers [7]. At first, the fundamental knowledge base was not sufficient for these ideas to be developed commercially, but as the field of DNA nanotechnology has matured, increasing attention has been given to exploitation. This article will discuss this change in the landscape, with reference to the broader context, intellectual property and patents, and the creation of new companies in the DNA nanotechnology space.

For the purposes of this work, DNA nanotechnology is defined as the branch of science and engineering that relates to nanoscale structures and devices that are made from DNA, and contain some artificial or designed elements. The DNA may be made chemically or biologically, and it may be functionalized or used as a scaffold for nanoparticles [8,9]. Data storage in DNA molecules *in vivo* should be excluded, as should most molecular biology and genetic engineering techniques. Inventions that involve ‘DNA and nanotechnology’ or nanotechnology-based systems for studying DNA (such as nanopore sequencers) should also be excluded. Aptamers should be included in some special cases, where the nature of the invention is incompletely specified by the sequences of the aptamers themselves. Although more than one type of DNA nanotechnology will be mentioned in this work, applications based on DNA origami will be emphasized [10], as this approach is well-defined, has been transformational for the field, and is a representative example of a design methodology.

In order for a technology to be commercially successful, it must address a particular need, which could not be addressed more effectively in another way. The technological basis for the product or idea constitutes intellectual property, and this can be protected by means of a patent [11]. If a patent is granted, the patentee has the exclusive right to exploit the invention for a specified period, and can take legal action if others attempt to do so. A patent application must provide a specification that fully describes the invention and what it does. The date on which the patent application is first filed is generally known as the priority date but under certain conditions, a patent application can claim the priority date of an earlier filing. The initial patent application is often filed with one particular country’s national office. Subsequently the application can be extended to other countries under the Patent Cooperation Treaty, according to which the priority date from the first national filing is carried over to the international phase, if the transition occurs within a certain time. Patent applications are assessed by examiners, who will consider such questions as whether the invention is novel and whether it involves an inventive step. In the UK, the Patents Act 1977 [12] also stipulates that the invention must have potential industrial applications, and specifies certain types of invention that are not patentable, including most software and some types of biological material. Globally, there is a long-running controversy over whether it should be possible to patent genes or not [13]. This will not be discussed further here, as it is not strictly relevant for most of the field of DNA nanotechnology.

About 18 months after filing, the patent application is published and made available to the general public. The patent application will not be released if it is withdrawn before the publication date. Once evaluation has been completed, the patent application may be granted. It usually takes years for a patent to be granted, and the patent will last for 20 years from the filing date. Prior public disclosure of an invention in journal articles or conference talks or similar will usually invalidate a patent application, although the reverse is not normally true. Reference should be made to an appropriate legal professional for definitive guidance on patents, intellectual property, or other legal matters related to the content of this article. The information presented here is correct to the best of the author’s knowledge, but no liability is accepted for any use to which it may be put.

There are a number of limitations and disadvantages to the patent system, such as the timescales, the costs, and the challenges of enforcing intellectual property rights. These issues are beyond the scope of the present work, in which patent applications are used as an indicator of intention to commercialize innovations in DNA nanotechnology. Evidence of actual commercialization is presented in the form of an overview of selected examples of spinouts/start-ups in the sector. The objective of a commercial company is to achieve profitability. Some critical considerations for starting a company include the timing, the existence of a potential market, and the competition from other companies [14]. It has been noted that the landscape for entrepreneurial activity in the biomedical field differs between countries [15]. Some of the many factors that contribute to the number of biomedical start-ups raising major funding are: the quantity of scientific output, the patent activity in the area, the presence of venture capital firms with an interest in the sector, and the human capital i.e. the availability of people with the correct skills and motivation [15].

As will be seen below, university research is an essential part of the innovation ecosystem. Commercialization is not the only link between academia and industry, but other types of academic engagement are not always easy to study as details of collaborative projects, consultancy arrangements and contract research are either not available publicly or not consolidated in a convenient format [16]. It is also important to note that for many innovations, commercialization would be impossible or impractical. Niche software tools are a classic example of this, and they are often released freely to the community via dedicated websites, servers or repositories, in accordance with the drive towards more open science [17]. There are several such software tools that are widely used in the field of DNA nanotechnology, and although they are available freely, it is instructive to consider the qualities they have that have enabled them to become standard in the field, displacing their competitors, and achieving near-total ‘market dominance’, as commercial products will seek to achieve the same.

## 2. Results

### 2.1. Analysis of Patent Applications Related to DNA Nanotechnology

Patent searches were carried out in November 2019 using the patent search service Espacenet, which is provided by the European Patent Office and is freely available to the general public [18]. It covers over 100 million patent documents [19], including those from other jurisdictions, such as the US, Japan, and China [20]. The title and abstract of published patent applications were searched for the strings given below. The obtained records were exported and analyzed. Some duplication of records arose due to the filing of applications in multiple jurisdictions or multiple filings for near-identical inventions, and these were removed.

Some ‘false hits’ were obtained in each search, relating to inventions that were not actually relevant to the search topic, and such records were also excluded. Examples include nanotechnology-based DNA sequencing approaches, or technologies that could be applied to many types of biomolecule. The decision was slightly subjective, and to avoid omissions the approach to exclusion was conservative, with the result that some ‘borderline’ results were included. Some ‘enabling technologies’ were included where the primary or only application lay in the area of DNA nanotechnology. The decision to include or exclude a patent was taken on the basis of the title, and/or information in the abstract and the cover page, where available. The full text was not generally used. Some patent applications were written in a language not familiar to the author of this paper, but these could still be included because a translation of the abstract was provided automatically by Espacenet. For some search strings an asterisk was appended, to ensure that variants on the wording would also be detected. For example ‘nanostructure*’ would cover ‘nanostructure’, ‘nanostructures’, and ‘nanostructured’.

As patents filed in 2018 would not necessarily have been published at the time of writing, data for that year would be incomplete and only patents filed prior to the end of 2017 were included. For all search strings, full details of included and excluded patents by title are given in Appendix A. Note that this data was not filtered for novelty or application status, and the list therefore includes numerous applications that will ultimately be rejected (i.e., not granted) for reasons, such as lack of novelty, inventive step, patentability. Only published applications are included, and the list therefore excludes patent applications withdrawn prior to publication.

The string ‘DNA nanotechnology’ was not an effective search string for Espacenet, as this is a word used to describe a field rather than explain what an invention is or does. Prior to filtering, this string initially produced 72 results but filtering for scope, duplication, and timeframe eliminated 40 of those. A search for ‘DNA nanostructure*’ produced 145 results prior to filtering. The removal of duplicates, and the results that were out of scope/timeframe or ambiguous, left 68 results. For these first two searches a little over half the results were removed by filtering. A search for ‘DNA origami’ initially produced 64 results, reduced to 41 after filtering. As would be expected, the two datasets overlapped. Interestingly, the original patent for the DNA origami method [21] did not feature in either of these datasets, because this patent referred to ‘nucleic acid nanostructures’ rather than ‘DNA’. This suggested another search string. A search for the broader term ‘nucleic acid nanostructure*’ produced 111 results, reduced to 70 after filtering.

A search was also performed using the advanced search for patents that contain the keyword ‘DNA’ in the title or abstract and are listed in category B82Y15/00 under the Cooperative Patent Classification scheme. This category is defined as ‘Nanotechnology for interacting, sensing or actuating, e.g., quantum dots as markers in protein assays or molecular motors’ [19]. This search produced 198 results, of which many were irrelevant. This dataset was therefore considered no further.

Note that the patents included in the analysis do not necessarily represent the comprehensive set of all patent applications related to the field of DNA nanotechnology. For example, some molecular motors made from DNA will be excluded. However, the records included are expected to be representative of the state of intellectual property in the field, and should indicate global trends.

For comparison with the scientific literature, Web of Science [22] was used. Searches were performed ‘by topic’ using the search strings ‘ “DNA nanotechnology” ’, ‘ “DNA nanostructure” ’ or ‘ “DNA origami” ’, where “ and ” were included in the search box. The timeframe considered was the same as that for the patent search (i.e., up to the end of 2017). Search results were analyzed directly using the Web of Science analysis tools. Filtering was not applied to remove any false results.

Figure 1 shows the trends for patent application filings (Figure 1a) and publication of papers (Figure 1b) for the search terms ‘DNA nanotechnology’, ‘DNA nanostructure(*)’ and ‘DNA origami’, where “ and “ were added as indicated in the labels. As explained above, ‘DNA nanotechnology’ is not an effective search string for Espacenet, giving only 32 results after filtering. All but 8 of those were filed in 2015 or later, which is clearly not an accurate reflection of the state of intellectual property in the field. In contrast, this is an effective search string for Web of Science, giving over 800 results up to the end of 2017, and this reflects the differences in use of language in patent applications and scientific papers. As Figure 1b indicates, the number of papers being published in this area is continuing to increase, with little sign of slowing down.

The Espacenet results for ‘DNA nanostructure*’ (after filtering, Figure 1a) show that the number of patent applications filed annually on this topic has been increasing gradually, following the publication of journal papers (Figure 1b) but with a lag time, as would be expected. The number of patent applications filed on ‘DNA origami’ remained low for a number of years, but there was a significant spike in 2017 (Figure 1a), for reasons that are not clear. In contrast, the number of papers published annually on this topic has been increasing steadily (Figure 1b), growing approximately 10-fold between 2007 and 2017.

For context, it is helpful to consider the number of research groups active in the area. The programme for a recent conference, ‘Functional DNA Nanotechnology 2018′, held in Rome, featured speakers from 22 different universities. Due to the location, European groups were represented particularly strongly. Lists of DNA nanotechnology research groups are sometimes to be found on personal websites, and one example is the list on the blog written by Arun Richard, which presently features around 100 entries [23]. The Web of Science results for papers published prior to the end of 2017 on the topic of “DNA nanotechnology” features 598 organizations, but many of these are associated with only one output. In this dataset, 161 organizations were associated with three or more publications, and the list of these organizations is given in Appendix A. It should be noted that research groups can move between universities, and there may be more than one research group at each institution.

Based on the available data, it is reasonable to suppose that 100–200 research groups globally are currently active in the area of DNA nanotechnology. Given that a Web of Science search yielded 169 items published on this topic in 2017, this would suggest an average of 1–2 outputs per group per year, if each output featured authors from only one group. As groups frequently collaborate, this is likely to be a serious underestimate.

As would be expected, the number of papers being published significantly exceeds the number of patents, as patent protection would be inappropriate for a large proportion of DNA nanotechnology research, including valuable studies on fundamental principles. For instance, the Web of Science search for DNA origami papers produced 228 results for 2017, twelve times the number of patent applications filed in the same year on the same topic. The data suggests that the fundamental science base (represented by the number of papers being published) is now sufficiently strong for attention to start turning to potential commercial applications (hence the increasing number of patent filings). This trend is likely to continue. At present, 13 records are available for DNA origami patents filed in 2018 or 2019, but as indicated above, this record is likely to be incomplete because applications are not typically published until 18 months after filing. An increase in patent applications after solidification of the knowledge base is a natural trend, and has been seen in other areas, such as graphene [24].

‘DNA origami’ patent applications were analyzed in more detail, with the results being shown in Figure 2 and given in full in Appendix A. In most cases, applications were filed by inventors of Chinese or American nationality (Figure 2a), where the nationality is either given in the patent document or assumed to be the same as that of the originating institution. Patent applications were also classified by source—academic, industrial or joint. This classification was based on the name of the applicant. For instance ‘X Univ’ or ‘Institute of Y’ were deemed to be academic institutions, while a name ending in ‘Co Ltd.’ was classified as I. The majority of patent applications in this area were filed by academic institutions (Figure 2b), in the form of universities or national research laboratories, which tends to suggest that major industrial players have yet to take a major interest in the topic, beyond participation in collaborative university projects.

A Google patents search indicated that some of the DNA origami patent applications had been withdrawn or discontinued, but many were still under evaluation. At least 11 had been granted in one or more jurisdictions, although in one of those cases, the status was described as ‘IP right cessation’. The number of inventors listed on the DNA origami patents varied from 1 to 9 (Figure 2c) and the most common number was 4. It is important to note that the threshold for inclusion as an inventor on a patent is usually deemed to be higher than that for inclusion as a co-author on a paper, as each inventor must make a contribution that is conceptual and not merely technical.

As will be seen from the Appendices, many of the patent applications examined here relate either to medical technologies, such as drug delivery and diagnostics, or fabrication techniques, such as lithography. These application areas are of particular commercial interest, with potential global markets worth billions of dollars.

### 2.2. DNA Nanotechnology Start-Ups and Spinouts

The companies discussed in this section were either known to the author previously or were found through Google searches. The case studies represent a sample rather than an exhaustive overview of commercial activity in the DNA nanotechnology sector. In all cases, DNA nanotechnology is a fundamental part of the company’s core business, which excludes companies that focus on sequencing (e.g., Illumina [25], Oxford Nanopore [26]), and synthesis (e.g., IDT [27], Twist Bioscience [28]). Also excluded are larger companies that may be carrying out some research on DNA nanotechnology as a side-line. Inevitably, the selection is biased towards those companies that operate primarily in English.

Table 1 provides four examples of companies operating in the DNA nanotechnology sector. All four of the companies selected have a well-defined product and core technology. Two of them (NuProbe [29] and GeniSphere [30]) focus on medical applications, an area with many potential competitors. For instance, even within the field of DNA nanotechnology, there are approaches for targeted drug delivery other than that promoted by GeniSphere.

In order to use diagnostics or therapeutics at the point-of-care, it is necessary to comply with various regulations and conduct extensive trials, involving significant expenditure of both time and money. Important indicators include approval by the FDA (Food and Drugs Administration, USA) and CE marking (Europe). Outside DNA nanotechnology, in the area of lab-on-a-chip diagnostics, it has been noted that the time between the founding of a company and the approval of its first product by a regulatory agency, such as the FDA or European Commission, can be 4–14 years [31]. Companies usually raise $10–50 million (US dollars) before their first point-of-care diagnostic is approved [31].

In the case of NuProbe, the assay kits they make are at present marked for research use only, not for use in diagnostics. For GeniSphere, rather than carrying out the research and development in house, the company works with others, such as biotech and pharmaceutical companies. This is a collaborative business model, and the symbiotic approach allows GeniSphere to focus on their core technology, providing materials, services and advice to their partners. Tilibit [32] and GATTAquant [33] also partner with others, but the partnerships listed in Table 1 appear to be primarily for distribution purposes, enabling the companies to reach a wider market. All four of these companies have or have had connections with universities. This is unsurprising in light of the observation made earlier that most of the patents in this sector have been filed by academic organizations. Note that the patents applicable to the core technology of the companies considered here did not all appear in the results of the Espacenet searches described earlier due to differences in terminology.

The youngest company listed in Table 1 was founded in 2016. There is less information available about more recent start-ups. FabricNano (UK) was founded in 2018 and aims to develop a DNA origami-based fabric-like material with embedded enzymes, with a view to speeding up production of biochemicals through the use of optimized reaction pathways [40]. The company, Nanovery (UK) was also founded in 2018 and aims to develop nanorobots that detect DNA in the bloodstream [41]. These companies are at a very early stage.

### 2.3. Software: Delivering Impact Without Commercialization

For some types of innovation, commercialization is not appropriate, and not all types of intellectual property can be patented. Software tools and programs are very difficult to patent, if not impossible, and other approaches are necessary to ensure that these innovations have an impact. A number of DNA nanotechnology software tools have been developed and made available to the community free of charge. It is instructive to consider some particularly successful examples. Selected cases are discussed below, noting the links between the software/services that are freely available and programs sold commercially.

The web-based ‘Nucleic Acid Package’ NUPACK [42] allows users to analyze the hybridization and secondary structure of DNA and RNA oligonucleotides. Its calculations are based on the thermodynamics of base interactions, including among other parameters the data derived by SantaLucia in Ref. [43]. NUPACK also offers some design functions for smaller DNA constructs, but does not take into account all the complexities of larger systems or the possibility of non-canonical DNA structures.

The introduction of the package cadnano revolutionized the design of DNA origami, making the process extremely intuitive [44]. Cadnano originated in an academic environment, but in recent years has also seen contributions from industrial collaborator Autodesk, Inc., which makes software for a wide range of Computer Aided Design operations, including tasks in mainstream mechanical engineering [45]. vHelix, a tool for free-form DNA nanostructure design [46], also works with Autodesk software, acting as a plugin for Maya.

It is possible to perform finite element modelling, to model the three-dimensional structure of DNA origami objects, by considering the mechanical properties of the molecules. A system, known as CanDo, has been developed for this [47]. CanDo is a web-based application that runs on a remote server, and the underlying platform is the finite element modelling software Abaqus. The use of Abaqus on the CanDO server is supported by Dassault Systemes, Inc., the company behind this program [48].

In the computer science community, it is common to place source code on the repository GitHub, and this has been used for various simulation and modelling tools, developed in the DNA nanotechnology community. Alternatively, code can be released through a dedicated website, as in the case of oxDNA, a package for modelling the dynamics of DNA using a course-graining approach [49].

Each of the programs or services mentioned above performs a particular function for which software was not previously available, and fulfils a niche role. Tools such as cadnano are used almost universally by practitioners in the field of DNA nanotechnology. However, the ‘market’ for these tools is still comparatively small. For instance, CanDo claims around 2500 unique users [48]. Many of these users would not have been prepared to pay for access to the resource, but even if fees had been taken from all of them for access, it is extremely unlikely that the funds raised would have covered the costs of developing and maintaining the system. Similar arguments apply to the other tools mentioned above. It is, therefore, clear that some of the technologies that have become critical for the development of DNA nanotechnology would not have been commercially viable, despite great success as free resources. However, there are important lessons to be learned from the near-total market dominance achieved by some of these tools. Firstly, they address an unmet need. Secondly, they are founded on a solid scientific base. Thirdly, the systems are intuitive, easy to use, and accompanied by clear documentation. Similar principles must be addressed by commercial products.

## 3. Discussion

An increasing number of patent applications is being filed in the area of DNA nanotechnology, and companies are being founded to commercialize intellectual property in this sector. It is to be expected that more entrepreneurial activity will follow, as the field has effectively come of age. External market conditions and trends can present both opportunities and threats; an example of this in another area is the effect of the ‘dot-com’ bubble on one silicon photonics company, with initial growth opportunities that later evaporated [50]. As the commercialization of DNA nanotechnology progresses, and products currently under development move up the scale of technology readiness levels, it is to be expected that the nature of the market will change. Initially, products may fill niche application areas, but in the longer term they may reach a wider base, with lower margins but higher volumes. This has also been predicted for graphene-based technologies [24].

In many areas, universities play a critical role in innovation. Many of the patents filed to date in the area of DNA origami originated in academic institutions, and three of the four companies profiled in this article grew out of university research groups, with the fourth mentioning that their first product was related to research carried out at Harvard. University intellectual property has also been extremely important in other areas, including silicon photonics [50], graphene [24], and lab-on-a-chip diagnostics [31]. However, academic and commercial practice are not always well-aligned, and the possibility exists for conflict, in addition to a wide range of legal, ethical, and political issues, as seen in recombinant DNA technology [51]. As DNA nanotechnology companies grow, they will inevitably carry out more research and development ‘in-house’, but various failures, acquisitions, and mergers are to be expected as the sector develops further, as seen in other areas.

Various obstacles must be overcome for the successful commercialization of DNA nanotechnology products. As in other sectors, the availability of funding for translation is critical, and bridging the gap between basic research and commercial application is challenging. This has been called the ‘valley of death’ or the ‘Darwinian Sea’, where the latter term is based on the image of ideas as big fish and little fish, swimming in a sea where only the fittest survive [52]. A supportive environment and strong innovation system will naturally be helpful for commercialization activities, and there are variations between countries in this regard [15], as alluded to earlier. Other issues mentioned above include access to a suitable talent pool, and the conflicts (real or perceived) that can arise between entrepreneurial and academic activities. The complexities and costs of the patent system may also deter some researchers from seeking intellectual property protection in this way, although this can be a very important aspect of commercialization.

Some issues are more sector-specific. For instance, it can be difficult to explain concisely to a non-specialist the difference between ‘genetic engineering’ and ‘engineering DNA for nanotechnology’. As many individuals unfamiliar with DNA nanotechnology *are* aware of the biological role of DNA, misunderstandings are possible. This presents a particular hazard for interactions with potential investors, patent attorneys and the public. Additionally, while regulations apply to therapeutics, it may not always be clear how they relate to DNA nanostructures and devices, and this may complicate matters.

The increased activity surrounding the commercialization of DNA nanotechnology is underpinned by a strengthened knowledge base. There is considerable potential for further expansion, because there are possible applications in areas, such as medicine and nanofabrication, which have huge target markets and numerous unsolved challenges. Developments in the field are likely to follow trends seen in other areas, and it is important to learn from successes and failures seen elsewhere. Some ideas that could not be effectively commercialized will, nevertheless, significantly influence progress, and it is essential for these to be disseminated effectively via the usual scientific channels, online servers and code repositories.

## Figures and Tables

**Figure 1 molecules-25-00377-f001:**
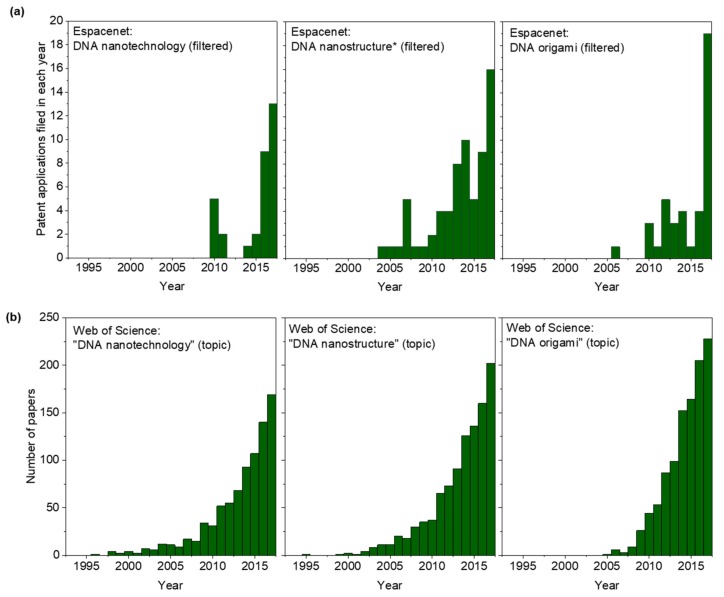
Patent applications and published papers in DNA nanotechnology. (**a**) Results from Espacenet searches for the indicated searches, after removal of duplicate and out-of-scope entries. (**b**) Web of Science results (unfiltered) for papers on indicated topics.

**Figure 2 molecules-25-00377-f002:**
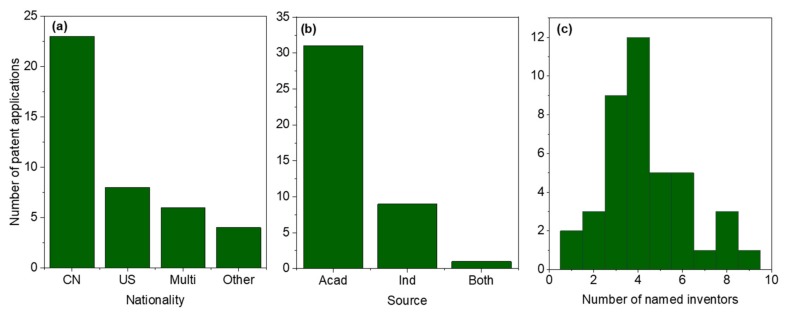
Further analysis of DNA origami patent applications. (**a**) Nationality of inventors. Taken directly from patents where given. Where inventor nationality was not given, the nationality of the originating institution was used instead. CN = China, US = United States, Multi = more than one nationality involved (all but one of these teams were partly American). (**b**) Type of originating institution – Acad = academic, i.e., university or national laboratory, Ind = industrial, Both = joint. (**c**) Number of inventors named on the patent applications.

**Table 1 molecules-25-00377-t001:** Examples of DNA nanotechnology-based start-ups and spin-outs. The content of this table is based on information provided in the websites of the companies themselves.

Company (and Year Founded)	Location (and University Connection, Where Applicable)	What It Does
NuProbe (2016) [29]	USA (Wyss Institute, Harvard & Rice University) & China	Core technology: Blocker Displacement Amplification (BDA) [34], enabling selective amplification of low abundance sequence variants
		Product: assay kits
		Problem addressed: identifying mutations associated with cancer, using DNA from blood or tumour samples, also working on infectious diseases.
		Market: researchers (not yet licensed for clinical/diagnostic use)
		State of development: available to order online.
		Other notes: Facilities in Boston, Houston and Shanghai. 6 board members, 2 additional Scientific Advisory Board members. Recently raised $11M in Series A funding
Tilibit nanosystems (2012) [32]	Germany (Technische Universität München)	Core technology: DNA origami [10]
		Product: origami materials and design/build/test services
		Problem addressed: economical supply of DNA origami materials, assistance with nanostructure preparation
		Market: individuals or organizations that wish to make DNA origami
		State of development: taking orders online. Customers include authors of Ref. [4].
		Other notes: partnerships with Eurofins Genomics [35] and IDT [27]
GATTAquant (2014) [33]	Germany (Technische Universität Braunschweig)	Core technology: DNA origami
		Product: DNA nanorulers [36]—origami objects carrying fluorophores with precisely defined separation
		Problem addressed: quantifying resolution of microscopes
		Market: users of super-resolution microscopy
		State of development: available to order online.
		Other notes: Partnership with Argolight [37] for distribution
GeniSphere (founded 1997, management buyout 2009, change in direction) [30]	USA (first product linked to Harvard researchers)	Core technology: 3DNA– construct consisting of a dsDNA core with double ss tails on both ends – can self-assemble into larger structures (dendrimers) [38]
		Product: early products included expression array detection kits, RNA labelling and amplification technologies; since 2009 buyout focus has been on drug delivery. Company provides materials, recommendations and services to other companies (biotech/pharma) about use of 3DNA to enhance their therapeutics
		Problem addressed: enhancing efficacy of therapeutics by targeted delivery of drugs or other active agents [39]
		Market: biotech/pharma companies
		Stage of development: various projects ongoing. 3DNA well-established, first patents in 1986.

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
