# Peer review of "The Business of DNA Nanotechnology: Commercialization of Origami and Other Technologies"

_molecules, 2020, doi:10.3390/molecules25020377_

Round 1
Reviewer 1 Report
K. D. gave a nice overview on the topic of DNA nanotech businesses. It sums a general picture on the status of DNA nanotech related commercialisation, but also can be a good reference for those who aimed to do any DNA nanotech related start-ups. I think it is a good contribution to the field thus would like to recommend to publish.
Q:
Is it also possible to include the data of how many DNA nanotech related research groups and research papers over the years?
Author Response
I thank the reviewer for the constructive comments. I am grateful to the reviewer for taking the time to read my paper and provide a review.
The reviewer asked ‘Is it also possible to include the data of how many DNA nanotech related research groups and research papers over the years?’. In response to this question, I have inserted the following text on page 4 of the manuscript (beginning at line 163 in the revised submission):
For context, it is helpful to consider the number of research groups active in the area. The programme for a recent conference, ‘Functional DNA nanotechnology 2018’, held in Rome, featured speakers from 22 different universities. Due to the location, European groups were represented particularly strongly. Lists of DNA nanotechnology research groups are sometimes to be found on personal websites, and one example is the list on the blog written by Arun Richard, which presently features around 100 entries (23). The Web of Science results for papers published prior to the end of 2017 on the topic of “DNA nanotechnology” features 598 organizations, but many of these are associated with only one output. In this dataset, 161 organizations were associated with three or more publications, and the list of these organizations is given in Supplementary Data 3. It should be noted that research groups can move between universities, and there may be more than research group at each institution.
Based on the available data, it is reasonable to suppose that 100-200 research groups globally are currently active in the area of DNA nanotechnology. Given that a Web of Science search yielded 169 items published on this topic in 2017, this would suggest an average of 1-2 outputs per group per year, if each output featured authors from only one group. As groups frequently collaborate, this is likely to be a serious underestimate.
As would be expected, the number of papers being published significantly exceeds the number of patents, because patent protection would be inappropriate for a large proportion of DNA nanotechnology research, including valuable studies on fundamental principles. For instance, the Web of Science search for DNA origami papers produced 228 results for 2017, twelve times the number of patent applications filed in the same year on the same topic.
Reviewer 2 Report
The article by K. Dunn is a comprehensive review of the current trend of DNA nanotechnology in regard to its commercialization. The reviewer recommends the article for publication. Some minor comments, mostly cosmetic, are provided below:
Main Text:
Some words can changed for more formal/academic equivalents such as: “more and more” (lines 12 and 281) and “obviously” (line 22) The abbreviation “etc” is used quiet extensively. In some instances, its use is not needed. For example, there is no need to write “etc.” in a list given within exempli gratia (e.g.). Also, most writing styles suggest punctuating the abbreviation “etc.” The numbering in Figures 1 and 2 can be set in the same style, e.g. (a) of the same font size on both figures.On the Supplementary Information:
Only the patent list for search string: “nucleic acid nanostructure*” is numbered. Numbering the tables used for other search strings could benefit the reader. Also, no footnotes to clarify the star in the search strings titles “DNA Nanostructure*” or “nucleic acid nanostructure*” are provided.
Author Response
I thank the reviewer for the constructive comments. I am grateful to the reviewer for taking the time to read my paper and provide a review.
Please find below a point-by-point response.
1. Some words can changed for more formal/academic equivalents such as: “more and more” (lines 12 and 281) and “obviously” (line 22)
I have changed the wording as follows:
‘More and more patent applications are being filed’ (originally lines 12 and 281) to ‘The number of patent application filings is increasing’ and ‘An increasing number of patent applications is being filed’. Removal of word ‘obviously’ in line 22I also reviewed the rest of the text and modified the phrasing in places. Examples of changes made include:
‘While’ to ‘although’ in line 35 ‘i.e.’ to ‘and hence’ in line 52 ‘make a profit’ to ‘achieve profitability’ in line 71 'players’ to ‘companies’ in line 73 ‘to err on the side of caution’ to ‘to avoid omissions the approach to exclusion was conservative, with the result that’ in line 104/105 ‘hits’ to results in lines 121-131 ‘the DNA nanotech space’ to ‘this sector’ in line 313This is not an exhaustive list. Line numbers refer to the revised submission and all changes have been tracked.
2. The abbreviation “etc” is used quite extensively. In some instances, its use is not needed. For example, there is no need to write “etc.” in a list given within exempli gratia (e.g.). Also, most writing styles suggest punctuating the abbreviation “etc.”
I have removed all instances of ‘etc’.
3. The numbering in Figures 1 and 2 can be set in the same style, e.g. (a) of the same font size on both figures.
I have corrected this.
On the Supplementary Information:
4. Only the patent list for search string: “nucleic acid nanostructure*” is numbered. Numbering the tables used for other search strings could benefit the reader.
I have done this.
5. Also, no footnotes to clarify the star in the search strings titles “DNA Nanostructure*” or “nucleic acid nanostructure*” are provided.
I have inserted the following statement into the Supplementary Information:
‘Here the asterisk is included in the search string to ensure that variants of the search term nanostructure are also included – such as ‘nanostructures’ or ‘nanostructured’.’
I also added a similar statement to the main text (line 111-113):
‘For some search strings an asterisk was appended, to ensure that variants on the wording would also be detected. For example ‘nanostructure*’ would cover ‘nanostructure’, ‘nanostructures’ and ‘nanostructured’.’
Reviewer 3 Report
In this paper, Katherine E. Dunn reviewed the commercial activity of DNA nanotechnology with an emphasis on DNA origami. A comprehensive analysis of patent applications in this field was performed by using search service Espacenet and Web of Science. The DNA nanotechnology start-ups and spinouts were discussed in terms of case studies. The software related to DNA nanotechnology was also included, which deliver impact without commercialization. It is revealed that the commercialization of DNA nanotechnology is emerging. The paper is well organized and written. The information provided in this manuscript is very instructive. I would suggest its publication after the following concerns are addressed.
1. Can the author discuss the challenges and difficulties in commercializing DNA nanotechnology?
2. All abbreviations should be well defined at the first use,e.g. “finite element modelling” (line 258).
3. It is better to include the original research papers in Bibliography so that readers lack of technological knowledge could easily trace them.
Author Response
I thank the reviewer for the constructive comments. I am grateful to the reviewer for taking the time to read my paper and provide a review.
Please find below a point-by-point response to the review.
Can the author discuss the challenges and difficulties in commercializing DNA nanotechnology?I have inserted the following text in the discussion:
'Various obstacles must be overcome for the successful commercialization of DNA nanotechnology products. As in other sectors, the availability of funding for translation is critical, and it is challenging to bridge the gap between basic research and commercial application. This has been called the ‘valley of death’ or the ‘Darwinian Sea’, the latter term being based on the image of ideas as big fish and little fish, swimming in a sea where only the fittest survive (52). A supportive environment and strong innovation system will naturally be helpful for commercialization activities, and there are variations between countries in this regard (15), as alluded to earlier. Other issues mentioned above include access to a suitable talent pool and the conflicts (real or perceived) that can arise between entrepreneurial and academic activities. The complexities and costs of the patent system may also deter some researchers from seeking intellectual property protection in this way, although this can be a very important aspect of commercialization.
'Some issues are more sector-specific. For instance, it can be difficult to explain concisely to a non-specialist the difference between ‘genetic engineering’ and ‘engineering DNA for nanotechnology’. As many individuals unfamiliar with DNA nanotechnology are aware of the biological role of DNA, misunderstandings are possible. This presents a particular hazard for interactions with potential investors, patent attorneys and the public. Additionally, while regulations apply to therapeutics, it may not always be clear how they relate to DNA nanostructures and devices, and this may complicate matters.'
All abbreviations should be well defined at the first use,g. “finite element modelling” (line 258).I have removed the abbreviation from line 258.
It is better to include the original research papers in Bibliography so that readers lack of technological knowledge could easily trace them.
I believe this point relates to the inclusion of company websites in the reference list. I have now added references to the original research papers that describe some of the companies’ key technologies, but I have left the website references in the list because some fundamental details about the businesses are not available in research papers.